# Peripheral Immunophenotype in IgG4-Related Disease and Its Association with Clinical Phenotypes and Disease Activity

**DOI:** 10.3390/cells12040670

**Published:** 2023-02-20

**Authors:** Eduardo Martín-Nares, Gabriela Hernández-Molina, Ángel A. Priego-Ranero, Isela Chan-Campos, Gladys S. Herrera-Noguera, Fidel López-Verdugo, Janette Furuzawa-Carballeda

**Affiliations:** 1Department of Immunology and Rheumatology, Instituto Nacional de Ciencias Médicas y Nutrición Salvador Zubirán, Vasco de Quiroga 15, Col. Belisario Dominguez Sección XVI, Mexico City 14080, Mexico; 2Department of Experimental Surgery, Instituto Nacional de Ciencias Médicas y Nutrición Salvador Zubirán, Vasco de Quiroga 15, Col. Belisario Dominguez Sección XVI, Mexico City 14080, Mexico

**Keywords:** immunoglobulin G4-related disease, phenotype, cytotoxic T lymphocytes, Th2 cells, T follicular helper cells

## Abstract

Diverse immune cell subsets have been described in IgG4-related disease (IgG4-RD). If there is a different immunophenotype according to clinical phenotype and activity status is not known. Levels of IL-4-, IL-13-, IL-5-, and IL-21-producing CD4^+^ T cells (Th2 subsets), CD4^+^ cytotoxic T lymphocytes (CD4^+^CTLs), T helper 9 cells, T follicular helper cells (Tfh; Tfh1/Tfh2/Tfh17/Tf regulatory [Tfr]), Foxp3^+^ regulatory T cells, Type 1 regulatory T cells (Tr1), T helper 3 regulatory cells (Th3), IL-10-producing regulatory B cells (Bregs), IL-10-expressing regulatory plasmacytoid dendritic (pDC IL-10^+^) cells, and M1 and M2 monocytes were determined by flow cytometry in 43 IgG4-RD patients and 12 controls. All immune subsets were higher in patients vs. controls. CD4^+^/IL-4^+^, CD4^+^/IL-5^+^, CD4^+^CTLs, Tfh2, Tfh17, Tfr, and M1 monocyte cell number was different among IgG4-RD clinical phenotypes. The pancreato-hepato-biliary phenotype was characterized by a higher CD4^+^CTLs, Tfh17, Tfh2, and Tfr and lower M1 cell number. An increased CD4^+^CTLs and Th3 cell number distinguished the head and neck-limited phenotype, while the retroperitoneal/aortic and Mikulicz/systemic phenotypes were characterized by increased Th2 subsets. Tfh17, Tr1, Th3, pDC, M1, and M2 monocytes were augmented in active patients. In summary, the clinical heterogeneity of IgG4-RD might be driven by the participation of different immunophenotypes and, consequently, by a different fibroinflammatory process.

## 1. Introduction

IgG4-related disease (IgG4-RD) is an emerging mass-forming fibroinflammatory condition characterized by tissue infiltration with IgG4^+^ plasma cells, affecting virtually any organ or anatomic site. However, there is a preference for specific organs such as major salivary glands, the orbit, the pancreas, and the retroperitoneum [1]. Furthermore, organ involvement tends to cluster in clinical phenotypes (pancreato-hepato-biliary, retroperitoneal/aortic, head and neck-limited, and Mikulicz/systemic) that differ in demographics, clinical and serological characteristics, and response to treatment. These clinical phenotypes were recently proposed and validated in external cohorts, and most patients with IgG4-RD fit into one [2,3]. 

Another way to classify patients is in proliferative and fibrotic phenotypes. In the former, glandular and epithelial organs are affected. It is usually multi-organic, has high serum IgG4 levels, has a good response to treatment, and is generally accompanied by atopy. In contrast, in the latter, anatomic sites are involved. It tends to be mono-organic, with normal serum IgG4 levels, and shows less treatment response [1]. The reason for this phenotypical variability is not known. However, it has been hypothesized that it may be due to genetic determinants or environmental factors [2].

On the other hand, the role of specific immune cell subsets in the pathogenesis of IgG4-RD is increasingly recognized. Studies have focused on CD4^+^ cytotoxic T lymphocytes (CD4^+^ CTLs), plasmablasts, T helper 2 (Th2) cells, T follicular helper (Tfh) cells, and M2 macrophages [4,5,6,7,8]. However, these studies have analyzed patients with only one manifestation of the disease (e.g., Mikulicz syndrome or autoimmune pancreatitis) or patients representing all phenotypes but without determining whether there are differences in the immunophenotype among them. 

In this study, we measured various immune cell subsets in patients with IgG4-RD: effector CD4^+^ T cells, Tfh cells, M1 and M2 monocytes, regulatory T, B, and plasmacytoid dendritic cells. We hypothesized that these subsets might be different compared to healthy controls and that different immunophenotypes might be encountered according to clinical phenotypes and activity status. 

## 2. Materials and Methods

### 2.1. Study Population

This was a cross-sectional study. We included patients with a diagnosis of IgG4-RD according to the Comprehensive Diagnostic Criteria for IgG4-RD and/or the Consensus Statement on Pathology [9,10] who attended a referral center in Mexico City from August 2018 to October 2019. The 2019 American College of Rheumatology (ACR)/European League Against Rheumatism (EULAR) classification criteria for IgG4-RD were retrospectively applied as they were published after the end of the recruitment of our cohort [11]. According to them, the patients were classified as definitive, probable, or atypical IgG4-RD, as suggested by Sanders et al. [12]. We excluded patients with concomitant diagnoses of another systemic autoimmune/inflammatory disease, active infection, or malignancy. As a control group, we recruited healthy subjects from the blood bank of our center. None had an autoimmune disease, a malignant disease, or a current/chronic infection. We obtained approval from the Institutional Review Board (IRE-2549-18-20-1), and the study complied with the Declaration of Helsinki. Patients and controls gave written informed consent. 

We retrospectively collected patient information from the clinical records, such as age at diagnosis, the number of organs involved, IgG4 serum levels at disease onset, biopsy results, and the use of glucocorticoids and immunosuppressive therapy at recruitment.

Patients were classified according to the clinical phenotypes described by Wallace et al. in pancreato-hepato-biliary, retroperitoneal/aortic, head and neck-limited, and Mikulicz/systemic phenotypes [2]. Patients were also classified according to the clinical phenotypes described by Zhang et al. in proliferative and fibrotic phenotypes [1]. Patients in the proliferative phenotype were those with involvement of glandular and epithelial organs (e.g., lacrimal and major salivary glands, pancreas, biliary tract, kidney, lung); patients in the fibrotic phenotype were those with involvement of extra-glandular sites or body regions rather than a specific organ (e.g., retroperitoneal fibrosis, mediastinal fibrosis, sclerosing mesenteritis) and those with pachymeningitis and Riedel’s thyroiditis [1]. We assessed the number of involved organs and the IgG4-RD Responder Index (IgG4-RD RI) at recruitment [13]. As a consensus definition of active and inactive disease is not available, we defined active disease as the presence of clinical signs and symptoms, laboratory abnormalities, or radiological findings attributable to IgG4-RD and an IgG4-RD RI ≥ 2. We defined inactive disease as the absence of clinical signs and symptoms, laboratory abnormalities, or radiological findings attributable to IgG4-RD and an IgG4-RD RI of 0, regardless of the use of immunosuppressive drugs. Patients were defined as atopic according to the definitions of the European Academy of Allergy and Clinical Immunology [14]. 

### 2.2. Peripheral Blood Mononuclear Cells Isolation and Activation

A venous blood sample (60 mL) was drawn from each subject to perform flow cytometry analysis. Peripheral blood mononuclear cells (PBMCs) were obtained by gradient centrifugation on Lymphoprep (Axis-Shield PoC AS, Oslo, Norway). The cell pellet was resuspended in 1 mL RPMI at 1–2 × 10^6^ cells/mL. Cell suspension was treated with 2 mL of a cell activation cocktail of phorbol-12 myristate 13-acetate (40.5 mM), ionomycin (669.3 mM) in DMSO (500×), and brefeldin A (BioLegend Inc., San Diego, CA, USA) for 6 h at 37 °C in a CO_2_ incubator.

### 2.3. T Memory Cell Purification

For magnetic depletion (negative selection) of naive T cells, CD8^+^ T cells, B cells, NK cells, γ/δ T cells, monocytes, DCs, granulocytes, platelets, and erythroid cells, PBMCs were incubated with a cocktail of biotinylated CD45RA, CD8, CD14, CD16, CD19, CD56, CD36, CD123, anti-TCRγ/δ, and CD235a (glycophorin A) antibodies. These cells were subsequently magnetically labeled with Anti-Biotin MicroBeads for depletion. Purity was assessed by fluorescence-activated cell sorting (FACS) staining for the T memory cell markers. Thus, anti-human CD4-PerCP and anti-human CD45RO-FITC monoclonal antibodies were used. This procedure usually yielded T memory-cell preparations >99% (Appendix A).

### 2.4. Flow Cytometry

PBMCs were incubated with 5 μL of Human TruStain FcXTM (BioLegend Inc., San Diego, CA, USA) per million cells in 100 μL PBS for 10 min, and then they were labeled with 2 µL of anti-human: (a) CD4 PerCP and CD14 APC antibodies; (b) CD4 PerCP and SLAMF7 APC antibodies; (c) ICOS FITC, Bcl6 PeCy7 and CXCR5 APC antibodies; (d) CD127 PE and CD86 APC antibodies; (e) CD163 PerCP and CD14 APC antibodies; (f) CD4 PerCP, CD127 PE, and CD25 APC antibodies; (g) CD4 PerCP, and CD25 APC antibodies; (h) CD163 PerCP, CD86 FITC, and CD14 APC antibodies; or (i) CD19 APC, CD24 FITC, and CD38 PerCP antibodies in separated tubes during 20 min at 37 °C in the dark. Cells were permeabilized with 200 µL of cytofix/cytoperm solution (BD Biosciences, NJ, USA) at 4 °C for 30 min. Intracellular staining was performed with an anti-human: (a) IL-13 FITC and IL-4 PE; (b) IL-5 PE; (c) IL-21 PE; (d) IL-1β PE and TGF-β1 ALEXA Fluor 488; (e) IL-17A PE; (f) IFN-**γ** PE; (g) IL-4 PE; (h) Foxp3 PE; (i) IL-9 PE; (j) TNF-α FITC; (k) IL-33 PE and TGF- β1 ALEXA Fluor 488; (l) Foxp3 ALEXA Fluor 488; or (m) IL-10 PE-labeled mouse monoclonal antibodies for 30 min at 4 °C in the dark. An electronic gate was made for live cells (FCSA vs. FCSH), then CD4^+^/CD14^−^ cells, CD4^+^/SLAMF7^+^ cells, Bcl6^+^/CXCR5^+^ cells, CD86^+^/CD127^+^ cells, CD163^+^ cells, CD4^+^/CD25^hi^ cells, CD19^+^/CD38^hi^/CD24^hi^ cells, and CD86^−^/CD14^+^/CD163^+^ cells. Results are expressed as the relative percentage of IL-13^+^, IL-4^+^, IL-5^+^, IL-9^+^, IL-21^+^, IL-1β^+^/TGF-β1^+^, IL-17A^+^, IFN-γ^+^, Foxp3^+^, TNF-α^+^, IL-10^+^, and IL-33^+^/TGF-β1^+^-expressing cells in each gate. As isotype control, IgG1-FITC/IgG1-PE/CD45-PeCy5 mouse IgG1 *kappa* (BD Tritest, BD Biosciences) was used to set the threshold and gates in the cytometer. We ran an unstained (autofluorescence control) and permeabilized PBMCs sample. Autofluorescence control was compared to single-stained cell positive controls to confirm that the stained cells were on the scale for each parameter. Moreover, BD Calibrate 3 beads were used to adjust instrument settings, set fluorescence compensation, and check instrument sensitivity (BD calibrates, BD Biosciences). Fluorescence minus one (FMO) controls were stained in parallel using the panel of antibodies with the sequential omission of intracellular antibodies. Finally, cell subsets were analyzed by flow cytometry with an Accuri C6 (BD Biosciences) in a blind manner regarding the clinical classification of the sample. A total of 500,000–1,000,000 events were recorded for each sample and analyzed with the FlowJo X software (Tree Star, Inc., Ashland, OR, USA). Antibodies’ characteristics are summarized in Appendix A. 

We determined the following immune cell subsets in patients and controls: IL-4-, IL-13-, IL-5-, IL-21-producing CD4^+^ T cells (Th2 subsets), CD4^+^CTLs, T helper 9 (Th9), Tfh (Tfh1/Tfh2/Tfh17/Tf regulatory [Tfr]), Foxp3^+^ regulatory T cells, Type 1 regulatory T cells (Tr1), T helper 3 regulatory cells (Th3), IL-10-producing regulatory B cells (Bregs), IL-10-expressing regulatory plasmacytoid dendritic (pDC IL-10^+^) cells, and M1 and M2 monocytes. The immunophenotype and staining panel of the evaluated cell subsets is shown in Appendix A. As our study was focused on circulating and not tissue cell subsets, we evaluated M1 and M2 monocytes, which have been proposed to mirror the M1/M2 macrophage polarization concept [15,16]. Representative plots from lymphocyte subpopulations are shown in Appendix A.

### 2.5. Assessment of Serum IgG1 and IgG4 Levels

Serum IgG1 and IgG4 levels were determined by turbidimetry analyzer (SPAPLUS, Freelite, The Binding Site, UK) at the time of recruitment, and normal values were set according to the manufacturer’s recommendations as follows: IgG1 normal range: 382.4–928.6 mg/dL; IgG4 normal range: 3.9–86.4 mg/dL.

### 2.6. Statistical Analysis

A descriptive analysis of the sample was performed. Dichotomous variables were expressed as absolute frequencies and continuous variables as means and standard deviations (SD) or medians and interquartile range (IQR) as appropriate. Comparison between means was made with Student’s *t*-test and between medians with Mann–Whitney U test. Dichotomous variables were analysed with the Chi-square test or Fisher exact test. We used the Kruskal–Wallis test for comparison among multiple groups and Dunn’s as a post hoc test. Correlations among variables were evaluated using Spearman’s test. A two-tailed *p* < 0.05 was considered statistically significant. All analyses were performed using the SPSS 20.0 and GraphPad Prism 8.3.0.

## 3. Results

A total of 43 IgG4-RD patients with a mean age of 52.3 ± 16.4 years were included; 21 (48.8%) were male. The cohort’s demographic, clinical, and serological characteristics are depicted in Table 1. Because a preliminary data analysis revealed significant differences between patients and controls, only 12 healthy subjects with a mean age of 40.6 ± 12.1 years were included; 5 (41.7%) were male. According to the 2019 ACR/EULAR classification criteria, 40 (93%) had definitive, 2 (4.7%) probable, and 1 (2.3%) atypical IgG4-RD. A total of 8 (18.6%) belonged to the pancreato-hepato-biliary, 7 (16.3%) to the retroperitoneal/aortic, 16 (37.2%) to the head and neck-limited, and 12 (27.9%) to the Mikulicz/systemic phenotypes, whereas 33 (76.7%) belonged to the proliferative and 10 (23.3%) to the fibrotic phenotypes. In total, 22 (51.2%) patients had active disease at recruitment, with a median number of organs involved of 2 (IQR: 2–5) and a median IgG4-RD RI of 6 (IQR: 4–12). All inactive patients had an IgG4-RD RI of 0. In total, 11 (25.6%) patients had a history of atopy. At study entry, median serum IgG1 was 715 mg/dL (IQR: 448–999), and 11 (25.6%) had high IgG1 serum levels, while median serum IgG4 was 66 mg/dL (IQR 31–115) and 15 (34.9%) had high IgG4 serum levels, 10 of them with active IgG4-RD.

A total of 27 (62.8%) patients were under immunosuppressive treatment at recruitment. A total of 17 were taking prednisone (median dose: 20 mg/day (IQR 5–35); median accumulated dose: 700 mg (IQR 145–1120); median duration: 28 days (IQR 14–140). In the sample, 19 were under immunosuppressor treatment; of those, 16 were taking azathioprine (median dose: 75 mg/day (IQR 50–112.5); median duration of 224 days (28–574), and 3 mycophenolate mofetil. None had received rituximab or methotrexate during their disease course. Of the 16 patients without immunosuppressive treatment, 10 had active disease. Of the 21 patients with inactive disease, 15 were taking immunosuppressive drugs.

### 3.1. Main Peripheral Subpopulations Altered in Patients with IgG4-RD Compared to Healthy Individuals

The patients with IgG4-RD presented statistically significant elevated levels of all subpopulations evaluated compared to healthy individuals (Table 2).

### 3.2. Main Peripheral Subpopulations Altered in Patients with IgG4-RD According to Clinical Phenotypes

Among the clinical phenotypes described by Wallace et al.,^2^ statistically significant differences in the number of the following cell subsets were found: CD4^+^ CTLs; (*p* = 0.04), Tfh (*p* = 0.04), Tfh2 (*p* = 0.03), Tfh17 (*p* = 0.006) and Tfr (*p* = 0.03), and a trend for IL-4-producing CD4 T cells (Th2; *p* = 0.05) and Th3 regulatory cells (*p* = 0.08) (Table 3). In a sensitivity analysis including only the 22 patients with active disease, only the difference in the number of Tfh17 cells remained statistically significant (*p* = 0.03). However, we observed differences in the concentrations of IL-4- (Th2; *p* = 0.03) and IL-5-producing CD4 T cells (Th2; *p* =0.01) and M1 monocytes (*p* = 0.04) (Table 4).

The pancreato-hepato-biliary phenotype was characterized by a higher CD4^+^CTLs, Tfh17, Tfh2, and Tfr and lower M1 cell number. An increased CD4^+^CTLs and Th3 cell number distinguished the head and neck-limited phenotype, while the retroperitoneal/aortic and Mikulicz/systemic phenotypes were characterized by increased Th2 subsets. 

There were no differences in cell subsets between the proliferative and the fibrotic phenotypes (Appendix A).

### 3.3. Immune Cell Subsets Differ According to Activity Status in IgG4-RD

Patients with active IgG4-RD showed higher numbers of Tfh17 (*p* = 0.03), Tr1 (*p* < 0.001), Th3 (*p* = 0.02), pDC IL-10^+^ (*p* = 0.009), M2 (*p* = 0.04), and M1 monocytes (*p* = 0.02), compared with patients with inactive disease (Figure 1; Appendix A). 

### 3.4. Immune cell Subsets According to the History of Atopy

When comparing the 11 patients with the 32 without a history of atopy, we found a lower count of Tfr cells (*p* = 0.04) in the atopic group and a trend for higher concentrations of IL-4- (Th2; *p* = 0.06) and IL-5-producing CD4 T cells (Th2; *p* =0.08) and M2 monocytes (*p* = 0.08) (Appendix A).

### 3.5. Immune Cell Subsets Did Not Differ in Patients under Immunosuppressive Treatment

We compared the 27 IgG4-RD patients under immunosuppressive treatment at study entry to the 16 patients without immunosuppressive therapy. There were no differences among immune cell subsets (Appendix A). In a sensitivity analysis comparing the patients without immunosuppressive treatment to the healthy subjects, we found similar findings as in the entire cohort (Appendix A). 

### 3.6. Correlations among Immune Cell Subsets and Clinical and Serological Variables

Immune cell subsets displayed correlations among them, as shown in Figure 2. Of note, IL-4-, IL-13-, IL-5- and IL-21-producing CD4 T cells positively correlated. The Tfh cells and their subtypes also positively correlated among them. The CD4^+^ CTLs cells positively correlated with regulatory cells and M1 and M2 monocytes. Finally, all regulatory cells also positively correlated among them. We did not find any correlation between immune cell subsets and IgG4 serum levels, the number of involved organs, or the IgG4-RD RI.

## 4. Discussion

IgG4-RD is an immune-mediated multi-organic fibroinflammatory disorder resulting from a dysregulated immune system. In this study, we found that patients with IgG4-RD have a higher cell concentration of a wide array of immune cell subsets than healthy subjects. We also demonstrated differences in those cell subsets according to clinical phenotype and activity status.

Regarding Th2 cells, they were first suggested to play a role in the pathogenesis of IgG4-RD in early works by Zen et al., in which Th2 cytokines were elevated in tissues from patients with mainly pancreato-hepato-biliary IgG4-RD manifestations [4]. Further studies corroborated that Th2 cells were elevated in blood and affected tissues, similar to our study [17,18,19]. In agreement with Mattoo et al., we also found higher Th2 concentrations in patients with a history of atopy [20]. Moreover, as Mattoo et al. described, CD4^+^ CTLs were markedly elevated in IgG4-RD compared to controls [21]. However, as previously reported, we did not find a correlation between serum IgG4 levels and the number of involved organs [22]. Recently, Pillai suggested that two different types of fibrosis coexist in IgG4-RD [23]. The first, termed “allergic fibrosis”, is orchestrated by Th2 cells and their cytokines. This type of fibrosis is demonstrated in the murine model for IgG4-RD, the LatY136F *knock-in* mice, which are characterized by an exuberant Th2 response and may reproduce the characteristic storiform fibrosis of IgG4-RD [24,25]. The second, termed “cytolytic fibrosis”, the triggering event is thought to be immune-mediated mesenchymal cell apoptosis by CTLs from both CD4^+^ and CD8^+^ lineages followed by a reparative process by macrophages, fibroblasts, and myofibroblasts [26]. Our study found that the immune cell subsets involved in both types of fibrosis, namely Th2 cells and CD4^+^ CTLs, were elevated compared to healthy subjects. Moreover, differences in the levels of those cell subsets among clinical phenotypes indicate that there may be a predominance of one fibrotic mechanism over the other. 

Recently, a subset of IL-9-producing T helper cells, termed Th9 cells, were implicated in inflammatory and atopic conditions and considered part of the Th2 immune response and IgE class-switch process [27,28]. Our study is the first to report that Th9 cells are elevated in IgG4-RD patients, which may contribute to the presence of atopy and elevation of IgE.

Further, studies have consistently reported that Tfh cells, particularly Tfh2, are increased in peripheral blood [18,29,30,31,32], affected organs [30,31,32], and secondary and tertiary lymphoid tissues [33], in patients with IgG4-RD and correlate with serum IgG4 levels and multi-organ involvement [29,30,31]. Tfh2 cells induce the differentiation of naïve B cells into plasmablasts and the subsequent production of IgG4 more efficiently than Tfh1 and Tfh17 cells [7]. In contrast with the studies mentioned above, we found that not only Tfh2 cells but also Tfh17 and Tfh1 cells were elevated in IgG4-RD patients. Furthermore, our findings suggest that Tfh17 cells may be pathogenic in IgG4-RD, as this cell subset was higher in active IgG4-RD. Although Grados et al. found high levels of Tfh17 cells in patients with IgG4-RD, it has been demonstrated that they do not efficiently induce IgG4 production [7,18]. Besides being a pro-inflammatory cytokine, IL-17 also has a profibrotic role in idiopathic pulmonary fibrosis and systemic sclerosis [34]. In this sense, the pathogenic role of Tfh17 cells and IL-17 in IgG4-RD needs further exploration. Tfr cells participate in the germinal center formation and class switch by producing IL-10 and TGF-β [35]. In agreement with Ito et al., this immune cell subset was elevated in patients with IgG4-RD [36].

On the other hand, the presence of regulatory cells was first described in the seminal works on the pathogenesis of IgG4-RD [4]. Since then, several studies have found high numbers of Tregs Foxp3^+^ cells in affected tissues [17,18,19,29,37,38]. Similarly, Sumimoto et al. demonstrated that Bregs CD19^+^CD24^hi^CD38^hi^ cells were elevated in patients with type 1 autoimmune pancreatitis compared to controls [39]. In contrast, Lin et al. reported that this immune cell subset was lower in IgG4-RD patients than in controls [40]. Other regulatory cell subsets have not been previously assessed in IgG4-RD (e.g., Tr1 and Th3 cells). Herein, we found that the four regulatory cells were elevated in our IgG4-RD population. Of note, the presence of high levels of IL-10-producing Tr1 cells in patients with the active disease was observed in this study. Tr1 cells are considered immunoregulatory cells since they secrete IL-10, which is essential in peripheral tolerance [41]. Tr1 cells may be acting as a compensatory anti-inflammatory mechanism. However, recent evidence in murine models of lung fibrosis suggests that IL-10 may be profibrotic, exacerbating Th2 responses, producing IL-4 and IL-13, the recruitment of fibrocytes, and M2 macrophage polarization [42,43]. Further studies are needed to clarify the role of IL-10 in IgG4-RD. 

Finally, macrophages have also been implicated in the pathogenesis of IgG4-RD. In particular, M2 macrophages contribute to the maintenance of Th2 responses, the infiltration by plasma cells, and the fibrotic process due to the production of APRIL, IL-10, CCL18, and IL-33 [8,44,45]. In addition, high levels of myeloid and pDCs have also been described [29,44]. Mechanistic studies have demonstrated that the activation of pDCs is implicated in the development of inflammation and fibrosis owing to the production of IFN-α and IL-33 [46]. Furthermore, IL-10-producing dendritic cells may have a tolerogenic role besides contributing to the differentiation of T cells into T regulatory and Tr1 cells and shutting down Th2 responses [47]. In this study, we found that patients with IgG4-RD have higher percentages of M1 and M2 monocytes and IL-10-producing pDC than healthy subjects. It is important to point out that M1 monocytes/macrophages have not previously been elevated in patients with IgG4-RD. M1 macrophages are considered pro-inflammatory due to their production of cytokines, namely TNF-α, IL-1α, IL-1β, IL-6, and IL-23, as well as reactive oxygen species. Nonetheless, there is evidence that TNF-α may be profibrotic in murine models of renal and lung fibrosis, while the evidence in IgG4-RD is limited [48,49]. Recently, Hong et al. reported that TNF-α produces tissue damage in IgG4-related sialadenitis due to autophagy [50].

Finally, we did not find correlations among the immune cell subsets and clinical parameters such as the number of involved organs, the IgG4-RD RI, or the IgG4 serum levels. The exact reason for our discordant findings cannot be explained with certainty; however, this discrepancy may be related to the different characteristics of the patient population in terms of ethnicity or activity status in each study. For instance, in the study by Cargill et al., including active and inactive IgG4-RD patients from the United Kingdom, the Tfh2 and Tfh17 cells correlated with the IgG4-RD RI and the Tfh2 cells correlated with the IgG4 levels. A French study including solely active patients found a correlation between Tfh2 cells with serum IgG4 levels and failed to find a correlation between the number of involved organs and the IgG4-RD RI [18,32]. 

Our study is not exempt from limitations. First, it was observational and focused on peripheral immune cell subsets, as we did not perform immunophenotyping of tissue cells or any functional or mechanistic study. Further studies are needed to address what happens at the tissue level or if the immunophenotype seen in patients with active disease, composed predominantly of immunoregulatory cells, is pathogenic or merely reflects a non-specific compensatory anti-inflammatory mechanism. Second, the cross-sectional design of our study limited us from establishing if our findings might change over time or with the start of treatment. Nevertheless, our results were similar in a sensitivity analysis only including untreated patients; if this immunophenotype remains the same during follow-up, it will need to be examined in future prospective studies. Third, our study population came from the same ethnic group, hampering the external validity of our results. Fourth, some phenotypes were underrepresented (e.g., retroperitoneal/aortic phenotype), and the small sample may have prevented us from finding more significant differences. Fifth, plasmablasts were not determined in our study. However, a previous study failed to find differences in their concentrations among clinical phenotypes [3]. Finally, the study population was small; however, considering IgG4-RD is a rare immune-mediated entity, the number of enrolled patients is significant, and the results are representative; although our control group was smaller, we observed striking differences compared with the IgG4-RD group.

Notwithstanding, our study is the first to analyze a complete immune cell repertoire related to the pathogenesis of IgG4-RD and to determine the differences among clinical phenotypes. Furthermore, for the first time, we have reported that some immune cell subsets are elevated in IgG4-RD, namely, Th9, Tr1, and Th3 cells and M1 and M2 monocytes. 

## 5. Conclusions

In conclusion, patients with IgG4-RD display a higher cell number of pro-inflammatory and profibrotic subsets than the healthy population. The clinical heterogeneity of IgG4-RD might be driven by the participation of different immune cell subsets and, consequently, a different fibroinflammatory process. Nevertheless, further prospective studies in larger cohorts are needed to confirm our findings and explore immune cell subsets’ usefulness as biomarkers for clinical phenotypes, activity, and response to treatment.

## Figures and Tables

**Figure 1 cells-12-00670-f001:**
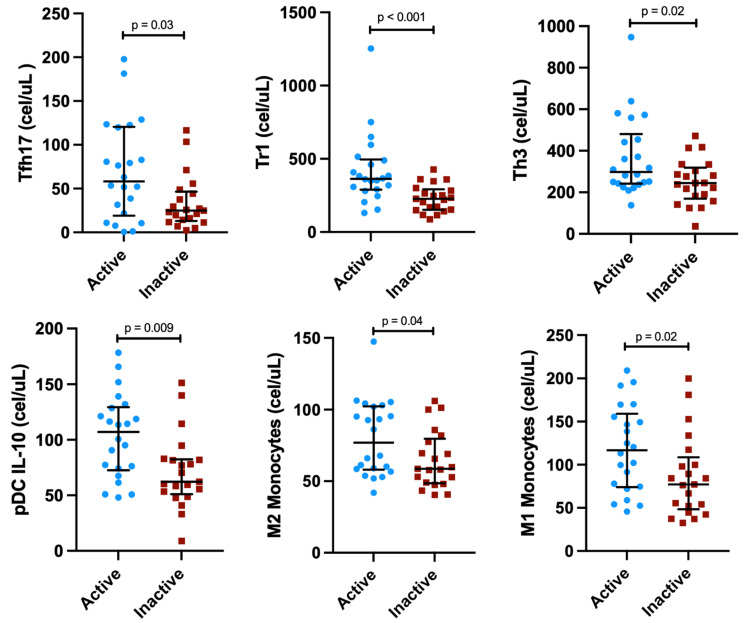
Concentrations of immune cell subsets differed between active vs. inactive IgG4-related disease patients.

**Figure 2 cells-12-00670-f002:**
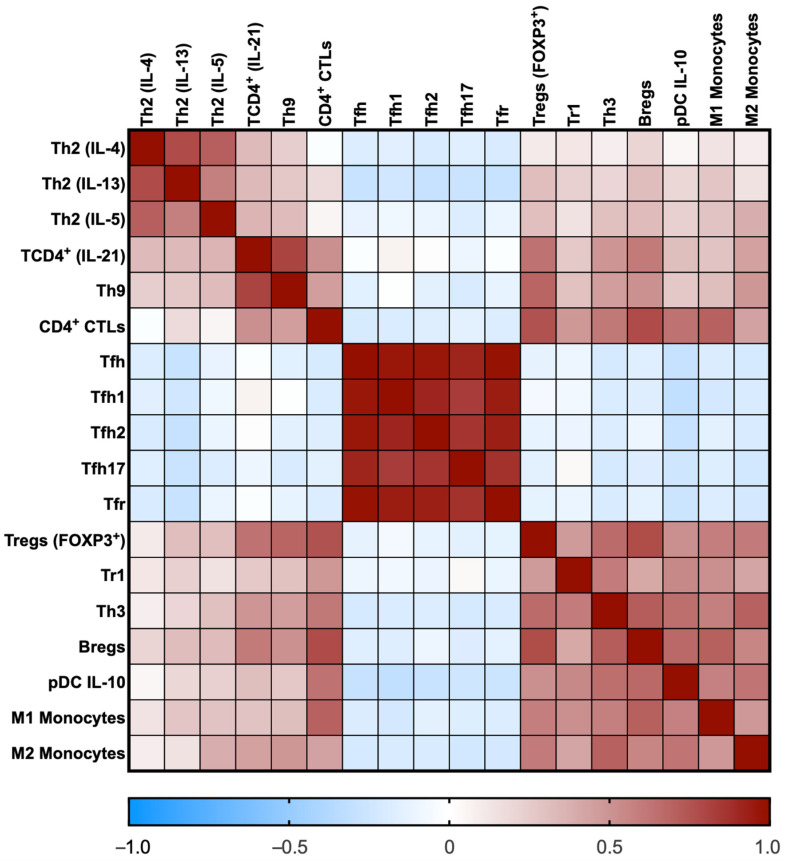
Heat map of the correlation coefficients among immune cell subsets. The rho coefficients are displayed as colors ranging from blue to red, as shown in the key (blue for negative and red for positive).

**Table 1 cells-12-00670-t001:** Demographic, clinical, and serological characteristics of IgG4-related disease patients.

	Overall Cohort (*n* = 43)	Pancreato-Hepato-Biliary (*n* = 8)	Retroperitoneal/Aortic (*n* = 7)	Head and Neck-Limited (*n* = 16)	Mikulicz/Systemic (*n* = 12)	Proliferative (*n* = 33)	Fibrotic (*n* = 10)	Active (*n* = 22)	Inactive (*n* = 21)
Age (years), mean ± SD	52.3 ± 16.4	57.7 ± 17.3	43.1 ± 16.9	51.7 ± 19.8	55 ± 7.2	52.9 ± 15.5	50.3 ± 19.7	50.6 ± 17.1	54.1 ± 15.7
Male, *n* (%)	21 (48.8)	6 (75)	5 (71.4)	3 (18.8)	7 (58.3)	16 (48.5)	5 (50)	10 (45.5)	11 (52.4)
Involvement of ≥ 3 organs (ever), *n* (%)	31 (72.1)	4 (50)	4 (57.1)	11 (68.8)	12 (100)	26 (78.8)	5 (50)	15 (68.2)	16 (76.2)
Involvement of 2 organs (ever), *n* (%)	8 (18.6)	3 (37.5)	2 (28.6)	3 (18.8)	0	5 (15.2)	3 (30)	4 (18.2)	4 (19)
Involvement of 1 organ (ever), *n* (%)	4 (9.3)	1 (12.5)	1 (14.3)	2 (12.5)	0	2 (6.1)	2 (20)	3 (13.6)	1 (4.8)
No. of organs involved (accrual), median (IQR)	4 (2–6)	2.5 (2–3)	3 (2–3.5)	4 (2–6)	8 (6–9)	5 (3–7)	2.5 (2–4)	4 (2–6)	4 (3–6)
Submandibular glands (ever), *n* (%)	23 (53.5)	3 (37.5)	2 (28.6)	8 (50)	10 (83.3)	20 (60.6)	3 (30)	10 (45.5)	13 (61.9)
Parotid glands (ever), *n* (%)	14 (32.6)	1 (12.5)	0	7 (43.8)	6 (50)	14 (42.4)	0	3 (13.6)	11 (52.4)
Lacrimal glands (ever), *n* (%)	18 (41.9)	0	1 (14.3)	9 (56.3)	8 (66.7)	17 (51.5)	1 (10)	8 (36.4)	10 (47.6)
Orbit (ever), *n* (%)	13 (30.2)	0	1 (14.3)	7 (43.8)	5 (41.7)	12 (36.4)	1 (10)	6 (27.3)	7 (33.3)
Paranasal sinus (ever), *n* (%)	14 (32.6)	1 (12.5)	0	7 (43.8)	6 (50)	13 (39.4)	1 (10)	6 (27.3)	8 (38.1)
Thyroid, *n* (%)	2 (4.7)	0	0	2 (12.5)	0	0	2 (20)	2 (9.1)	0
Lymph nodes (ever), *n* (%)	18 (41.9)	1 (12.5)	1 (14.3)	6 (37.5)	10 (83.3)	16 (48.5)	2 (20)	9 (40.9)	9 (42.9)
Lung, (ever) *n* (%)	11 (25.6)	0	0	2 (12.5)	9 (75)	11 (33.3)	0	5 (22.7)	6 (28.6)
Mediastinal fibrosis (ever), *n* (%)	3 (6.9)	0	2 (28.6)	0	1 (8.3)	1 (3)	2 (20)	3 (13.6)	0
Pancreas (ever), *n* (%)	15 (34.9)	6 (75)	1 (14.3)	1 (6.3)	7 (58.3)	14 (42.4)	1 (10)	6 (27.3)	9 (42.9)
Biliary tract (ever), *n* (%)	12 (27.9)	7 (87.5)	0	0	5 (41.6)	12 (36.4)	0	6 (27.3)	6 (28.6)
Gallbladder (ever), *n* (%)	1 (2.3)	1 (12.5)	0	0	0	1 (3)	0	0	1 (4.8)
Mesentery (ever), *n* (%)	3 (6.9)	0	2 (28.6)	0	1 (8.3)	1 (3)	2 (20)	1 (4.5)	2 (9.5)
Kidney (ever), *n* (%)	10 (23.3)	0	0	0	10 (83.3)	10 (30.3)	0	5 (22.7)	5 (23.8)
Retroperitoneal fibrosis (ever), *n* (%)	5 (11.6)	0	5 (71.4)	0	0	0	5 (50)	3 (13.6)	2 (9.5)
Aorta (ever), *n* (%)	1 (2.3)	0	0	0	1 (8.3)	1 (3)	0	0	1 (4.8)
Prostate (ever), *n* (%)	3 (6.9)	0	0	0	3 (25)	3 (9.1)	0	2 (9.1)	1 (4.8)
High IgG4 levels (ever), *n*+/*n* (%)	28/42 (66.7)	6/7 (85.7)	4 (57.1)	7 (43.8)	11 (91.7)	23/32 (71.9)	5 (50)	14 (63.6)	14/20 (70)
High IgG4 levels at recruitment, *n* (%)	15 (34.9)	3 (37.5)	2 (28.6)	3 (18.8)	7 (58.3)	13 (39.4)	2 (20)	10 (45.5)	5 (23.8)
High IgG1 levels at recruitment, *n* (%)	11 (25.6)	2 (25)	2 (28.6)	5 (31.3)	2 (16.7)	9 (27.3)	2 (20)	7 (31.8)	4 (19)
Immunosuppressive treatment at recruitment, *n* (%)	27 (62.8)	4 (50)	5 (71.4)	11 (68.8)	7 (58.3)	21 (63.6)	6 (60)	12 (54.5)	15 (71.4)
Active disease at recruitment, *n* (%)	22 (51.2)	3 (37.5)	4 (57.1)	9 (56.3)	6 (50)	15 (45.5)	7 (70)	-	-

IQR: interquartile range; SD: standard deviation.

**Table 2 cells-12-00670-t002:** Immune cell subsets in IgG4-related disease and healthy subjects.

	IgG4-Related Disease (*n* = 43)	Controls (*n* = 12)	*p*
CD4^+^/CD14^−^/IL-4^+^ (cell/µL)	289.98 (155.1–495.7)	89.23 (66.3–111.9)	<0.001
CD4^+^/CD14^−^/IL-13^+^ (cell/µL)	399.9 (251.2–616.4)	89.4 (72.7–152.3)	<0.001
CD4^+^/CD14^−^/IL-5^+^ (cell/µL)	361 (201.1–503.6)	1.38 (0.93–1.82)	<0.001
CD4^+^/CD14^−^/IL-21^+^ (cell/µL)	380.2 (280.7–665.1)	131.8 (95.4–153)	<0.001
CD4^+^/CD14^−^/IL-9^+^ (cell/µL)	405 (244.3–630.7)	94 (64.1–217.1)	<0.001
CD4^+^/SLAMF7^+^/IL-1β^+^/TGF-β1^+^ (cell/µL)	1774.8 (1397.1–2307)	454.7 (398.6–630.8)	<0.001
Tfh (CD4^+^/CXCR5^+^/ICOS^+^/Bcl6^+^) (cell/µL)	179.3 (113–333.6)	20.6 (19.3–21.7)	<0.001
Tfh1 (cell/µL)	53.65 (21.6–76.5)	5.6 (4.5–6.1)	<0.001
Tfh2 (cell/µL)	47.8 (28.6–78.7)	4.8 (4.5–5.5)	<0.001
Tfh17 (cell/µL)	37.5 (15.2–77.8)	4.6 (3.9–6)	0.001
Tfr (cell/µL)	48.6 (29.9–82.1)	5.6 (5.3–6.2)	<0.001
CD4^+^/CD127^−^/CD25^hi^/Foxp3^+^ (Tregs; cell/µL)	354.5 (268.2–474.2)	244.1 (158.7–291.5)	0.002
CD4^+^/CD25^low^/Foxp3^−^/IL-10^+^ (Tr1; cell/µL)	292.1 (204.5–374.6)	140.8 (80.4–192.2)	<0.001
CD4^+^/CD25^−^/Foxp3^−^/TGF-β1^+^ (Th3; cell/µL)	276.5 (221.4–365.6)	98.1 (70.6–137)	<0.001
CD19^+^/CD24^hi^/CD38^hi^/IL-10^+^ (Bregs; cell/µL)	95.2 (62.2–127.7)	56.1 (44.5–70.9)	0.001
CD86^−^/CD163^hi^/IL-10^+^ (pDC; cell/µL)	78.1 (59.1–115.5)	63.1 (49.8–77.1)	0.04
CD163^+^/TGF-β1^+^/IL-33^+^ (M2 Monocytes; cell/µL)	65.7 (53.5–94.2)	37.9 (29.8–49.3)	<0.001
CD86^+^/CD127^+^/TNF-α^+^ (M1 Monocytes; cell/µL)	91.5 (57.2–142.6)	43.7 (39.8–49.5)	<0.001

Results are presented as medians of cells per microliter for each immune cell subset (interquartile range).

**Table 3 cells-12-00670-t003:** Immune cell subsets in clinical phenotypes of IgG4-related disease.

	Pancreato-Hepato-Biliary (*n* = 8)	Retroperitoneal/Aortic (*n* = 7)	Head and Neck-Limited (*n* = 16)	Mikulicz/Systemic (*n* = 12)	*p*
CD4^+^/CD14^−^/IL-4^+^ (cell/µL)	186.2 (98.5–221.6)	346.6 (305.1–620.6)	257.6 (104.7–527.3)	391.7 (313.9–541)	0.05
CD4^+^/CD14^−^/IL-13^+^ (cell/µL)	279.7 (197.3–327.7)	623.7 (336.7–836.8)	419.3 (229.8–626)	449.5 (262.9–599.5)	0.26
CD4^+^/CD14^−^/IL-5^+^ (cell/µL)	201.2 (154.9–261.3)	361 (328–486.6)	318.6 (191.5–450-6)	480.2 (305.4–646.6)	0.12
CD4^+^/CD14^−^/IL-21^+^ (cell/µL)	340.6 (248.3–571.9)	380.3 (332.9–544.6)	506.6 (373.9–737.9)	300.9 (192.5–457.8)	0.15
CD4^+^/CD14^−^/IL-9^+^ (cell/µL)	538.3 (471.4–764.2)	405 (311.6–612.3)	407.6 (251.3–617.8)	320.2 (149–500.5)	0.27
CD4^+^/SLAMF7^+^/IL-1β^+^/TGF-β1^+^ (cell/µL)	2139.5 (1698.3–2616.6)	1354.5 (1033.8–2125.4)	2023.4 (1614.9–3204.2)	1564 (850.8–1897.5)	0.04
Tfh (CD4^+^/CXCR5^+^/ICOS^+^/Bcl6^+^) (cell/µL)	363.6 (233.6–496.8)	174.1 (26.7–201.4)	201.5 (136.4–315.5)	132.6 (55.7–251.3)	0.04 *
Tfh1 (cell/µL)	86.1 (51.3–111.6)	37 (6.7–59.9)	57.8 (34.5–68.5)	47.5 (11.7–69.3)	0.13
Tfh2 (cell/µL)	84.7 (61.3–127.7)	39.2 (6.5–45.8)	54.7 (36.1–72.9)	32 (14.4–65.8)	0.03 *
Tfh17 (cell/µL)	91.3 (57.7–122.6)	48.8 (7.3–54)	40.6 (22.2–81.8)	16.9 (7.7–34.5)	0.006 *
Tfr (cell/µL)	101.5 (63.4–115.4)	34.5 (6.2–48.8)	53.9 (31.7–72.5)	34.7 (14.6–63.9)	0.03 *
CD4^+^/CD127^−^/CD25^hi^/Foxp3^+^ (Tregs; cell/µL)	453.5 (331.1–519.5)	321.5 (305.7–381.9)	412.1 (278.8–478.1)	268.2 (203.6–420.6)	0.17
CD4^+^/CD25^low^/Foxp3^−^/IL-10^+^ (Tr1; cell/µL)	243.8 (163.4–389.2)	360.2 (245.9–392.2)	327.7 (239.8–359.9)	249.1 (142.2–396.7)	0.55
CD4^+^/CD25^−^/Foxp3^−^/TGF-β1^+^ (Th3; cell/µL)	263.2 (231.7–302.2)	237.9 (199.8–245.6)	347.7 (266.3–500.8)	241.3 (173.6–321.8)	0.08
CD19^+^/CD24^hi^/CD38^hi^/IL-10^+^ (Bregs; cell/µL)	102.7 (88.3–120.4)	70.2 (58.4–107.5)	118 (81.4–148.3)	63.6 (46.5–123.9)	0.10
CD86^−^/CD163^hi^/IL-10^+^ (pDC; cell/µL)	94.9 (58.4–107.5)	67.6 (55.6–97.7)	86 (75.5–141.6)	58.8 (50.9–84.3)	0.11
CD163^+^/TGF-β1^+^/IL-33^+^ (M2 Monocytes; cell/µL)	60.5 (58.4–98.3)	66 (52.5–90.4)	73.5 (54.8–92.9)	58.8 (50.9–84.3)	0.77
CD86^+^/CD127^+^/TNF-α^+^ (M1 Monocytes; cell/µL)	83.4 (53.3–149.2)	113.2 (60.3–147.2)	95.6 (77.2–127.2)	84.9 (43.5–148)	0.81

Results are presented as medians of cells per microliter for each immune cell subset (interquartile range). * *p* < 0.05 for the comparison between pancreato-hepato-biliary and Mikulicz/systemic phenotype.

**Table 4 cells-12-00670-t004:** Immune cell subsets in clinical phenotypes in active IgG4-related disease.

	Pancreato-Hepato-Biliary (*n* = 3)	Retroperitoneal/Aortic (*n* = 4)	Head and Neck Limited (*n* = 9)	Mikulicz/Systemic (*n* = 6)	*p*
CD4^+^/CD14^−^/IL-4^+^ (cell/µL)	174.6 (130–193.7)	308.2 (171.8–648.6)	106.6 (95.7–234.3)	499.6 (379.1–918)	0.03 ***
CD4^+^/CD14^−^/IL-13^+^ (cell/µL)	338.5 (291.6–415.3)	702 (247.3–1073)	281.5 (142.4–489)	599.5 (480.5–884)	0.25
CD4^+^/CD14^−^/IL-5^+^ (cell/µL)	123.6 (99–200.6)	328 (201.6–437.1)	237.9 (141.3–363.7)	623.2 (505–962.5)	0.01 **
CD4^+^/CD14^−^/IL-21^+^ (cell/µL)	222 (180.8–248.3)	518.3 (332.9–945.7)	390.1 (305.8–471.4)	432.4 (287–1099)	0.14
CD4^+^/CD14^−^/IL-9^+^ (cell/µL)	417.7 (313.7–481.9)	524.5 (311.6–756.3)	262.8 (238.5–393.5)	489.7 (282–1040.3)	0.41
CD4^+^/SLAMF7^+^/IL-1β^+^/TGF-β1^+^ (cell/µL)	1865.5 (1033.8–3182.2)	1946.8 (1773.9–3229.4)	1946.8 (1773.9–3229.4)	1616.6 (1439.6–1771)	0.32
Tfh (CD4^+^/CXCR5^+^/ICOS^+^/Bcl6^+^) (cell/µL)	513.1 (430–520.5)	100.3 (15.8–413.5)	304.8 (166.4–341.1)	165 (83.3–324.8)	0.14
Tfh1 (cell/µL)	106.3 (94.6–113.5)	21.5 (3.5–84.3)	61.6 (44.4–69.3)	48.2 (16.6–84.9)	0.17
Tfh2 (cell/µL)	125.4 (104.2–136.4)	28.7 (4.1–81.7)	67.8 (38.8–82.7)	49.3 (28.8–121.9)	0.14
Tfh17 (cell/µL)	128.7 (103.9–155.1)	29.9 (4.4–125)	80.7 (53.4–119.9)	26.6 (10.5–38.6)	0.03 **
Tfr (cell/µL)	114.9 (108.2–115.42)	20.1 (3.4–122.4)	70.5 (34.8–84.9)	43.9 (21.1–79.2)	0.17
CD4^+^/CD127^−^/CD25^hi^/Foxp3^+^ (Tregs; cell/µL)	354.5 (308.6–441.8)	381.9 (281.9–557.2)	468.9 (276.7–481.8)	332.3 (212.7–503.1)	0.95
CD4^+^/CD25^low^/Foxp3^−^/IL-10^+^ (Tr1; cell/µL)	460.2 (389.5–528)	395.2 (371.2–830.8)	351 (292.2–384.8)	285.9 (153.7–488.7)	0.40
CD4^+^/CD25^−^/Foxp3^−^/TGF-β1^+^ (Th3; cell/µL)	249.86 (193.6–284)	245.5 (225.8–351.3)	442 (286.2–573)	295.3 (237.7–371.2)	0.20
CD19^+^/CD24^hi^/CD38^hi^/IL-10^+^ (Bregs; cell/µL)	81.4 (62.8–105.8)	76.4 (47–120.4)	124.1 (84.8–144.5)	89.9 (57.9–131)	0.45
CD86^−^/CD163^hi^/IL-10^+^ (pDC; cell/µL)	91.2 (78.2–97.9)	90.6 (59.3–114)	121.2 (90.5–151.8)	96.3 (50.5–128.6)	0.19
CD163^+^/TGF-β1^+^/IL-33^+^ (M2 Monocytes; cell/µL)	95.5 (78.4–100.4)	80.6 (59–99.7)	86.1 (56.7–93.2)	59.6 (58.5–103)	0.78
CD86^+^/CD127^+^/TNF-α^+^ (M1 Monocytes; cell/µL)	52.5 (49–53.3)	147.2 (119–180.7)	102 (91.5–170.1)	142.6 (72–149.3)	0.04 *

Results are presented as medians of cells per microliter for each immune cell subset (interquartile range). * *p* < 0.05 for the comparison between pancreato-hepato-biliary and retroperitoneal/aortic phenotype. ** *p* < 0.05 for the comparison between pancreato-hepato-biliary and Mikulicz/systemic phenotype. *** *p* < 0.05 for comparing head and neck-limited and Mikulicz/systemic phenotype.

## Data Availability

The datasets used during the current study are available from the corresponding author upon reasonable request.

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
