# Peer review of "Peripheral Immunophenotype in IgG4-Related Disease and Its Association with Clinical Phenotypes and Disease Activity"

_cells, 2023, doi:10.3390/cells12040670_

Round 1
Reviewer 1 Report
In the manuscript entitled “Peripheral immunophenotype in IgG4-related disease and its association with clinical phenotypes and disease activity”, the authors evaluated a wide array of immune cell subsets in patients with IgG4-RD: effector CD4+ T cells, Tfh subsets, M1 and M2 monocytes, regulatory T, B, and plasmacytoid dendritic cells. patients with IgG4-RD displayed a higher cell number of pro-inflammatory and profibrotic subsets compared to the healthy population. Patients with active IgG4-RD showed higher numbers of Tfh17, Tr1, Th3, pDC IL-10+, M2, and M1 monocytes compared with patients with inactive disease. However, they did not find any correlation between immune cell subsets and IgG4 serum levels, the number of involved organs, or the IgG4-RD RI. The following are major and minor points that need to be addressed:
In Materials and Methods
1. line 100, the volume of 60 ml blood needs revision
2. The markers assessed and the fluorochrome used need revision. Some are missing from the supp. table S1 or missing in the text, for example:
a. ICOS, IL17A, and IL-10 FITC are not mentioned in the description of the procedure but are mentioned in the supp. table S1
b. CD38 mab is labeled with PeCy5 in the procedure description, but in the supp. table S1 labeled with PerCP
c. In line 135, you mentioned assessing the level of IL-33+/TGF-β + expressing cells but you didn’t mention using them in the same tube in the procedure as you did for TGF-β and IL-β in line 127
d. CD45RO FITC was not mentioned in supp. Table S1
3. The fluorochromes described in the flow cytometry dot plots are not matching with that in supp. Table S1. For example compare CD4, Bcl6, and CD38 in the supp. Table S1 with those in the plots in the supp. figures
4. The fluorochrome used with anti-CD45RO was FITC as was described in line 114 and for IL-13 was also FITC (supp. table S1). How did you assess CD4+/CD45RO+/CD14-/IL-13+ using anti-CD45RO and anti-IL13 in the same tube?
Similarly, The fluorochrome used with anti-ICOS was FITC (supp. Table S 1), How did you assess Tfh subsets and Tfr using anti-CD45RO and anti-ICOS in the same tube?
Also, anti-CD127 and FoxP3 were labeled with PE. How did you assess you assess Tregs?
5. The number of healthy controls is deficient despite being collected from healthy blood donors. They should be increased to give more reliable results
In the results
1. The percentages in table 1 need revision. For example:
a. the percentage of the biliary tract in Mikulicz/systemic is 41.7%, not 27.9%
b. The submandibular glands in the proliferative phenotype represent 60.6%, not 60%
c. Also, lymph nodes in the active disease represent 40.95, not 40%
2. Correct the count of pDC in supp. Table (S3) from 1107.1 to 107.1
3. Some typos should be corrected like Cel/ul
Reviewer 2 Report
The present paper written by Martín-Nares et al, is aimed to establish the differences in cellular subsets among the different phenotype and activity status in patients affected with IgG4-RD. It is an important issue, regarding that such entity has increasing interest among physicians of different specialties. This work pretends to evaluate if there are any differences on in the efficacy of corticosteroids in the treatment of this disease.
Major comments
Given that there are not standardized data regarding the normal range of cellular population, probably the number of patients and controls included should be higher.
As authors stated, the work is aimed in evaluated peripheral cells, and not study the population subsets of affected tissue, that could have provide more valuable results.
The fact that 62.7% of patients were under immunosuppressive treatment at the recruitment it is an important issue, because probably has an impact on the proportion of subset cells.
Minor comments
Reviewer 3 Report
The authors did a descriptive cross-sectional study from a single referral center in Mexico City over a 14-month period. They included 43 IgG4-RD patients according to the 2019ACR/EULAR criteria with exclusion of any other autoimmune disease, active/chronic infection, and malignancy. Twelve healthy control patients were included for the study as well. They examined a complete immune cell repertoire in the peripheral blood sample that has been implicated in the pathogenesis of IgG4-RD and analyzed any differences among the known clinical phenotypes.
The results from this study appear novel and provoking new questions, despite the limitations of descriptive cross-sectional study. Although the study size is reasonable (in fact, surprisingly high for 14-month period), it is still a small study and would require further investigation from a larger cohort. The lack of tissue examination in this study is another area to be addressed in a future study if possible.
